# Enhancing Privacy in Multimodal Federated Learning with Information Theory

**Tianzhe Xiao**[1], **Yichen Li**[1], **Yining Qi**[1], **Yi Liu**[2], **Wei Wang**[2],
**Haozhao Wang**[1*], **Yi Wang**[2], **Ruixuan Li**[1]

[1]School of Computer Science and Technology, Huazhong University of Science and Technology,
Wuhan, China [2]Chongqing Ant Consumer Finance Co., Ltd, Ant Group, Chongqing, China
{d202381469, ycli0204, qiyining, hz_wang, rxli}@hust.edu.cn

## Abstract

Multimodal federated learning (MMFL) has gained increasing popularity due to its ability to leverage the correlation between various modalities, meanwhile preserving data privacy for different clients. However, recent studies show that correlation between modalities increase the vulnerability of federated learning against Gradient Inversion Attack (GIA). The complicated situation of MMFL privacy preserving can be summarized as follows: 1) different modality transmits different amounts of information, thus requires various protection strength; 2) correlation between modalities should be taken into account. This paper introduces an information theory perspective to analyze the leaked privacy in process of MMFL, and tries to propose a more reasonable protection method **Sec-MMFL** based on assessing different information leakage possibilities of each modality by conditional mutual information and adjust the corresponding protection strength. Moreover, we use mutual information to reduce the cross-modality information leakage in MMFL. Experiments have proven that our method can bring more balanced and comprehensive protection at an acceptable cost.

## 1 Introduction

Federated Learning (FL) has emerged as a fundamental paradigm that enables collaborative model training among multiple parties via parameter aggregation without sharing private datasets [31, 27, 14]. Owing to its privacy-preserving and communication-efficient nature, FL has been widely deployed in diverse applications, including smart healthcare [1, 38] , financial analysis [50, 30, 5] and recommendation system[22]. Some of the major research directions in FL also include efficient aggregation[45], communication compression[20, 19, 18], continual learning[24, 25, 21, 23], and knowledge distillation[44].

However, FL is not immune to security threats. A notable attack, Gradient Inversion Attack (GIA), aims to infer sensitive information from the shared model updates (gradients) [26]. Malicious participants can exploit this to reconstruct private data or infer its properties, thereby breaching local privacy [59, 13, 56].

To counter this risk, several defense methods have been proposed in FL [46, 15]. LDP-Fed [43] optimizes local differential privacy (LDP) for FL, ensuring a lightweight and quantifiable privacy measure. In [7], regularization and sparsification techniques are employed to alleviate performance degradation with user-level DP. FedDPA [51] explores differential privacy in personalized FL through dynamic Fisher personalization and adaptive constraints, while PrivateRec [28] focuses on federated recommendation, aiming to improve model utility under DP guarantees.

Previous FL approaches are mainly trained using uni-modal data, yet real-world data is often multi-modal. For example, videos typically come with audio tracks and text subtitles, and internet content

like social media posts and news articles often blend text, images, videos, and audio. Multi-modal federated learning (MMFL) tasks such as image annotation, visual question answering, and image-text retrieval leverage complementary information from different modalities, resulting in a global model that outperforms uni-modal counterparts [9, 6, 54].

While existing defense approaches may work well for uni-modal FL, they struggle in MMFL for two key reasons:

*First, applying the same protection strength across different modalities is inappropriate.* Different modalities have distinct representation formats and reconstruction challenges. For instance, image data is typically represented as high-dimensional pixel matrices (e.g., a 224×224×3 image contains roughly 150,000 parameters) with continuous, smoothly varying features that require coordinated recovery of multiple pixels to restore semantic content. In contrast, text data is represented by discrete words or embeddings (e.g., Word2Vec or BERT), where recovering key words usually suffices to retain most semantic information. As shown

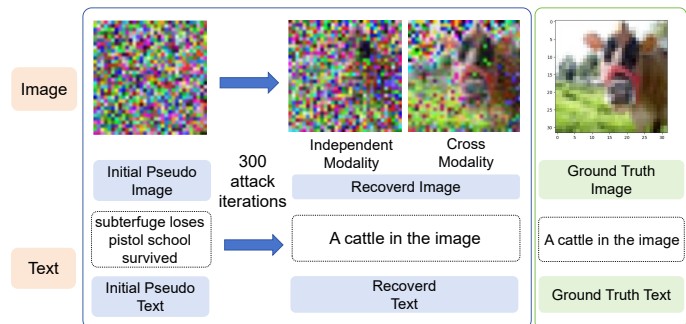

Figure 1: Different recovery rates of data from various modalities under attack. For each independent modality, starting from random pseudo data, after 300 rounds under the same setting, the recovered text perfectly matches the original, while the recovered image remains blurry. However, when we let the information of each modality interact to carry out cross modality attacks, under the same attack iterations, the images can be recovered more similar using the information recovered from text modality.

in Figure 1, after 300 rounds of attack starting from random pseudo data, the recovered text is completely consistent with the original, whereas the recovered image remains blurry. Uniform protection can therefore either over-distort one modality—hindering the fusion of useful multi-modal information and reducing accuracy—or under-protect another, leaving gradients vulnerable to leakage. This discrepancy in data representation demands a nuanced approach to protection. By carefully calibrating the level of noise or perturbation for each modality, one can better balance privacy preservation and performance. In practice, determining the optimal protection strength requires a deep understanding of each modality's inherent characteristics.

*Second, inter-modal data correlation can intensify the impact of GIAs.* In MMFL, training data for different modalities are paired, leading to a "barrel effect" where the most vulnerable modality is breached first, accelerating leakage across all modalities. Models typically align features from different modalities (e.g., images and text) into a common space to capture their relationships. This alignment increases cross-modal semantic associations in the gradients, enabling attackers to infer more precise information and even deduce the content of one modality from another[29]. Furthermore, the close coupling of modalities means that even a minor breach in one channel may compromise the security of the entire system. Understanding and mitigating these risks is essential to developing robust privacy-preserving techniques in MMFL. Such cross-modal alignment thus broadens the scope of privacy leakage.

To address these challenges, we propose **Sec-MMFL**, which secures MMFL both effectively and efficiently. We begin by analyzing the causes and flow of training information leakage in MMFL. Integrating information theory, we propose a method to measure leakage risk across modalities, allowing for modality-specific protection strengths. Additionally, we study the impact of inter-modal correlations on privacy leakage and develop methods to mitigate the heightened risk of cross-modal gradient inversion attacks without sacrificing training accuracy.

- We analyze the causes of training information leakage in MMFL and propose a novel information-theoretic framework to quantify leakage risk across modalities, allowing us to assign appropriate protection based on each modality's vulnerability to gradient inversion attacks.

- We introduce **Sec-MMFL**, an adaptive protection framework that assigns modality-specific protection strengths to mitigate privacy risks from inter-modal correlations while maintaining high model performance.

- Extensive experiments on benchmark datasets like CIFAR-10, CIFAR-100, Hateful-Memes and CrisisMMD demonstrate that **Sec-MMFL** outperforms traditional methods under equivalent privacy guarantees, effectively balancing privacy preservation and model utility.

## 2 Related Work

### 2.1 Multimodal Federated Learning

Multimodal federated learning (MMFL) has emerged as a promising paradigm that synergizes the privacy-preserving nature of federated learning [8, 16] with the representational power of multimodal learning [39, 34]. This framework has demonstrated significant potential in real-world applications ranging from affective computing [11, 33] to distributed healthcare systems [42, 36], particularly through its deployment in IoT sensor networks [35, 37, 58]. The fusion methodology in MMFL typically operates through three principal approaches: early fusion that combines raw feature representations, late fusion that aggregates model outputs, and hybrid strategies that integrate both paradigms [2, 10, 4, 12]. Our investigation focuses on the fundamental dichotomy between early and late fusion to elucidate their distinct impacts on privacy-preserving mechanisms.

### 2.2 Gradient Inversion Attack

The security vulnerabilities of federated learning systems have been extensively documented, with gradient inversion attacks [49] representing one of the most potent threats to data privacy. These attacks exploit the mathematical properties of shared gradients to reconstruct private training data through iterative optimization of pseudo-inputs [59]. Recent methodological advancements have significantly enhanced attack efficacy through innovations in prior-informed initialization [17], ground-truth label recovery [57], and specialized regularization techniques [13, 53], enabling successful breaches even against large-batch training scenarios and complex architectures like vision transformers [53]. This evolving threat landscape underscores the critical need for robust defensive countermeasures.

### 2.3 Privacy Protection in Distributed Learning

Contemporary privacy-preserving techniques employ multi-layered protection strategies. Differential privacy mechanisms inject calibrated noise into gradient updates [43, 47], and similar noise perturbation strategies have also been explored to defend against backdoor attacks[52]. Secure aggregation protocols enable encrypted parameter aggregation without exposing individual updates [3]. Homomorphic encryption further extends protection by permitting computations on ciphertexts [55]. Although recent work by [41] proposes an information-theoretic framework for privacy leakage assessment in unimodal settings, the unique challenges posed by multimodal data interactions in MMFL remain largely unaddressed, highlighting a crucial gap in existing literature.

## 3 Preliminaries and Problem Statement

### 3.1 FL Procedures.

We aim to collaboratively train a global model for $K$ total clients in FL. We consider each client $k$ can only access to his local private dataset $D_k := \{x_i, y_i\}$, where $x_i$ is the $i$-th input data sample and $y_i \in \{1, 2, \cdots, C\}$ is the corresponding label of $x_i$ with $C$ classes. Specifically in MMFL, $x_i = \{x_i^1, x_i^2, ..., x_i^m\}$, where $x_i^m$ is the $i$-th input data sample of the $m$-th modality. The global dataset is considered as the composition of all local datasets $D = \sum_{k=1}^{K} D_k$. The objective of the FL learning system is to learn a global model $w$ that minimizes the total empirical loss over the entire dataset $D$:

$$\min_{w} \mathcal{L}(w) := \sum_{k=1}^{K} \frac{|D_k|}{|D|} \mathcal{L}_k(w), \text{ where } \mathcal{L}_k(w) = \frac{1}{|D_k|} \sum_{i=1}^{|D_k|} \mathcal{L}_{CE}(w; x_i, y_i), \qquad (1)$$

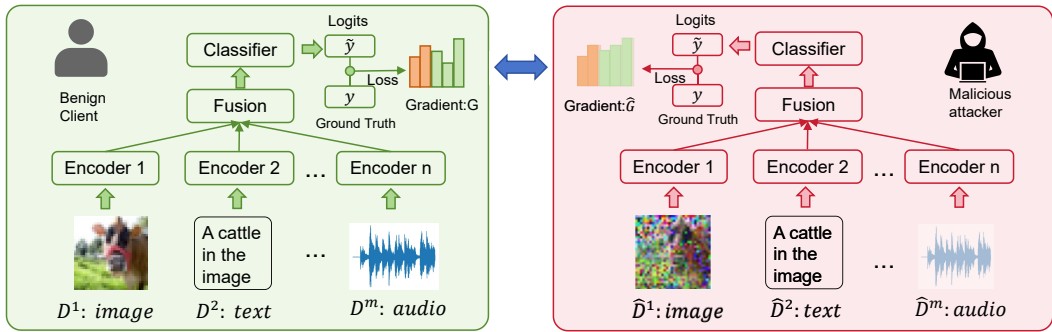

Figure 2: The information flow of GIA in MMFL. Information from the benign client's training data flows into the embedding after being processed by different modality encoders. Following fusion and classification, it is then incorporated into the gradient through the loss function. A malicious attacker can recover the information from the different modalities of the training data by eavesdropping on the gradient.

where $\mathcal{L}_k(w)$ is the local loss in the $k$-th client and $\mathcal{L}_{CE}$ is the cross-entropy loss function that measures the difference between the prediction and the ground truth labels.

Each client updates its model parameters using gradient descent:

$$w_{t+1}^k = w_t - \eta \nabla \mathcal{L}_k(w_t), \tag{2}$$

where $\eta$ is the learning rate, and $\nabla \mathcal{L}_k(w)$ represents the gradient of the loss function with respect to the parameters, denoted as $G^k$.

The central server aggregates the gradients from all clients to update the global model parameters:

$$w_{t+1} = w_t - \eta_t G_t, \text{ where } G_t = \sum_{k=1}^{K} \frac{|D_k|}{|D|} G_t^k, \tag{3}$$

## 3.2 GIA Procedures.

GIA exploit the information encoded in the gradients to reconstruct the original data $D$. The attacker's process can be summarized as follows:

1. Initialize a random guess $\hat{D}$ for the data.

2. Iteratively refine $\hat{D}$ by minimizing the difference between the gradients computed from $\hat{D}$ and the observed gradients $G$:

$$\hat{D} \leftarrow \hat{D} - \eta \nabla_{\hat{D}} \left( \|G - \nabla \mathcal{L}(\hat{D}; w)\|^2 \right) \tag{4}$$

3. The refined $\hat{D}$ converges to an estimate of the original data.

This process highlights how gradients can leak sensitive information, and the way attackers obtain information about training data from it. The whole process of data of each modality in MMFL flowing to the attacker through GIA is shown in Figure 2.

## 3.3 Information-Theoretic Preliminaries

The conditional entropy of $X$ given $Y$ represents the remaining uncertainty of $X$ after observing $Y$:

$$H(X|Y) = - \sum_{x \in X, y \in Y} p(x, y) \log p(x|y) \tag{5}$$

The mutual information between $X$ and $Y$ quantifies the amount of information shared between the two random variables:

$$I(X; Y) = H(X) - H(X|Y) \tag{6}$$

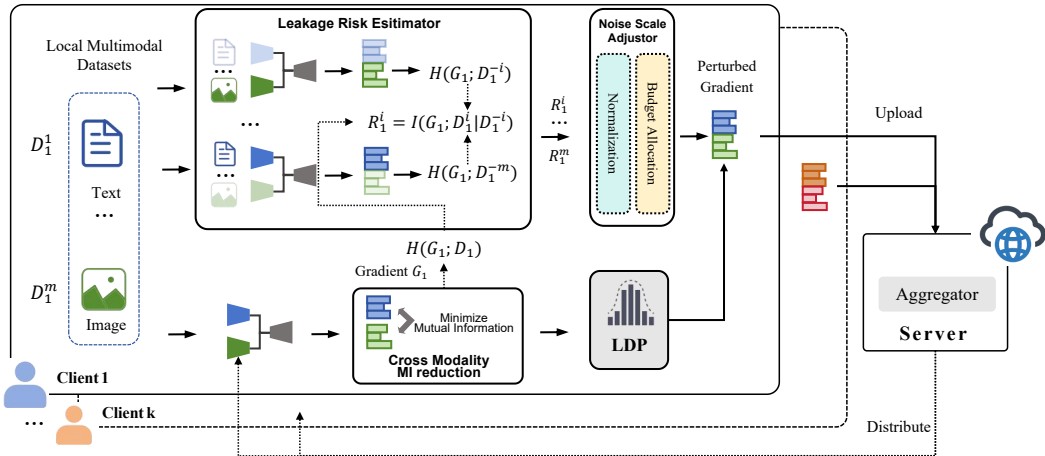

Figure 3: The overall architecture of Sec-MMFL. After calculating the gradient of the multimodal model, the local training data is processed by the Leakage Risk Estimator to compute the conditional mutual information between the original data of each modality and the gradient. This is used to assess the risk of data leakage for each modality through the gradient. Based on this, the Noise Scale Adjustment module adjusts the privacy budget assigned to each modality, achieving a better balance between privacy protection and model effectiveness. Meanwhile, the Cross Modality MI Reduction module reduces the mutual information between gradients of different modalities, thereby mitigating cross-modal privacy leakage risks.

## 3.4 Local Differential Privacy

A perturbation algorithm $M$ satisfies $(\epsilon, \delta)$-Local Differential Privacy ($(\epsilon, \delta)$-LDP) if, for any pair of adjacent datasets $D$ and $D'$, and for all possible output subsets $S$, the following inequality holds:

$$Pr[\mathcal{M}(D) \in S] \leq e^{\epsilon} Pr[\mathcal{M}(D') \in S] + \delta \tag{7}$$

where $\epsilon$ is the privacy budget of $M$, which quantifies the privacy protection level, and $\delta$ is the probability of the privacy guarantee being violated. A smaller value for $\epsilon$ indicates a smaller gap between two probabilities and thus a stronger privacy.

## 4 Proposed Method

The workflow of the proposed framework is shown in Algorithm 1, and Fig. 3 illustrates the Sec-MMFL approach.

### 4.1 Leakage Risk Assessment

While assessing the information leakage of the gradient through GIA and applying corresponding protection before sending gradients to servers and evaluating the similarity between reconstructed data and real data offers direct insights, it is computationally expensive due to its iterative nature, often requiring thousands of iterations to converge. Mutual information serves as a more efficient approach to quantify the information leakage in scenarios like MMFL; it can not only measure the information dependence between the raw data and the gradient but also consider the correlation between the modalities.

It is essential to formalize the description of the complete information leakage channel. In the whole process, we consider the dataset $D$ as the sender and the estimated data $\hat{D}$ obtained through the attacker's iterative optimization as the receiver. The complete information leakage channel can be described as a process from the data $D$ to the gradients $G$, and subsequently from the gradients $G$ to the estimated data $\hat{D}$.

**Algorithm 1:** Sec-MMFL

**Input :** $T$: number of communication rounds; $K$: number of clients; $\eta$: learning rate; $\{D_k\}_{k=1}^{K}$: distributed datasets; $w$: initial model parameters; $\lambda$: hyperparameter for MI reduction.
**Output :** Trained global model parameters $w$

1 Initialize the parameter $w$;
2 **for** $t = 1$ *to* $T$ **do**
3      Server randomly selects client subset $S_t$ and sends $w$ to them;
4      **for** *each selected client* $k \in S_t$ **do**
5          compute the gradient $G$ of model;
6          **Leakage Risk Assessment**:
7          **for** *each modality* **do**
8              Assess the risk $R_i$ of leakage of data from $i$-th modality using conditional mutual information with 8;
9          **end**
10         **Noise Scale Adjustment**:
11         **for** *each modality* **do**
12             Adjust the scale of noise added for $i$-th modality with 9,10,11and 12;
13         **end**
14         **Cross Modality MI reduction**:
15         Reduce the mutual information between gradient of encoders across modalities with 13 and 14;
16         Update local model parameters;
17         Send the updated model $w^k$ back to the server;
18      **end**
19      **At server side:**;
20      Aggregate the received models $w^k$ from clients in $S_t$ to update the global model $w$;
21 **end**

However, the channel from gradients $G$ back to the estimated data $\hat{D}$ is subject to the attacker's attack methods and prior knowledge, which we cannot fully evaluate and control. Therefore, we focus on the controllable part of the channel, which is from the data $D$ to the gradients $G$.

When we evaluate the risk of data leakage from each modality, if we directly calculate mutual information between $D^i$ and $G$, it reflects the direct relationship between data and gradients without any additional conditions. But in MMFL, the influence of data from one modality on the gradient may be influenced or supplemented by data from other modalities; therefore, we choose conditional mutual information to evaluate the information sent by raw data $D^i$ and received by gradients $G$ in the channel which reflects the risk of leakage $R_i$ of data from that modality:

$$R_i = I(G; D^i \mid D^{-i}) = H(G \mid D^{-i}) - H(G \mid D) \tag{8}$$

where $H(G \mid D^{-i})$ is the entropy of the gradients given all modalities except $D^i$, and $H(G \mid D)$ is the entropy of the gradients given all modalities.

The conditional mutual information $I(G; D^i \mid D^{-i})$ serves as a key metric for quantifying the privacy leakage of each modality. A higher value indicates that more information about $D^i$ is encoded in the gradients $G$, even when the data from other modalities $D^{-i}$ is known. This can be particularly concerning if an adversary can access these gradients, as it indicates the potential to reconstruct or infer sensitive attributes of $D^i$.

### 4.1.1 Noise Scale Adjustment

To balance privacy preservation and model utility in Sec-MMFL, we allocate noise scales $\sigma_i$ for each modality using Rényi Differential Privacy (RDP), leveraging an information-theoretic approach to achieve tighter privacy accounting under composition. The leakage risk for modality $i$ is given by $R_i$, as defined in Eq. (8). We compute a normalized risk weight for each modality via softmax:

$$w_i = \frac{\exp(-R_i)}{\sum_{t=1}^{M} \exp(-R_t)} \tag{9}$$

where $M$ is the total number of modalities. To ensure higher-risk modalities receive stronger protection, we define a scaling factor $s_i = 1/\sqrt{w_i}$, which amplifies noise for modalities with larger $R_i$ (and thus smaller $w_i$). The noise scale for modality $i$ is then set as:

$$\sigma_i = c \cdot s_i, \quad \text{with} \quad s_i = \frac{1}{\sqrt{w_i}} \tag{10}$$

The global scaling factor $c$ is determined via binary search to satisfy the target $(\epsilon_{\text{target}}, \delta)$-DP guarantee. The total RDP cost at order $\alpha$ is:

$$\rho_{\text{total}}(\alpha) = \sum_{i=1}^{M} \rho_i(\alpha), \quad \text{where} \quad \rho_i(\alpha) = \text{compute\_rdp}(q_i, c \cdot s_i, S_i, \alpha) \tag{11}$$

Here, $q_i$ is the sampling rate, $S_i$ is the gradient clipping norm, and $\alpha > 1$ is the Rényi order for modality $i$. This summation is valid for each fixed $\alpha > 1$ due to the additive composition property of RDP under independent Gaussian mechanisms. We enforce the privacy budget by ensuring:

$$\epsilon = \min_{\alpha > 1} \left[ \rho_{\text{total}}(\alpha) + \frac{\log(1/\delta)}{\alpha - 1} \right] \leq \epsilon_{\text{target}}, \quad \text{for a fixed } \delta \tag{12}$$

This optimization ensures the cumulative RDP across all modalities satisfies the target privacy level. The resulting $\sigma_i$ values are applied to each modality's PrivacyEngine in Opacus, enabling efficient privacy-preserving training with minimal utility loss.

### 4.2 Cross Modality MI reduction

In MMFL, the close relationship between different modalities can lead to increased privacy leakage risks. For instance, if images and text pairs are closely related, and the text typically describes the image, then the ability to infer the text information could accelerate and increase the probability of the text being extracted by an attacker. Due to the interconnection between different modalities, the mutual information between their encoders' gradients is also high. To address this issue, we propose a method that, in addition to calculating the task loss related to the accuracy (e.g., cross-entropy loss), includes an additional term to reduce the mutual information between the gradients of different modality encoders before propagating the gradients back for model updates.

Let $L_{\text{task}}$ be the task-specific loss, such as cross-entropy loss, and $L_{\text{MI}}$ be the loss term designed to reduce the mutual information between the gradients of different modality encoders. The total loss function $L_{\text{total}}$ is then given by:

$$L_{\text{total}} = L_{\text{task}} + \lambda L_{\text{MI}} \tag{13}$$

where $\lambda$ is a hyperparameter that balances the importance of reducing mutual information against the task-specific loss.

The mutual information reduction loss $L_{\text{MI}}$ can be defined based on the gradients of the different modality encoders $G^1, G^2, \ldots, G^m$ as follows:

$$L_{\text{MI}} = \sum_{i=1}^{m-1} \sum_{j=i+1}^{m} I(G^i; G^j) \tag{14}$$

where $I(G^i; G^j)$ denotes the mutual information between the gradients of the $i$-th and $j$-th modality encoders.

By minimizing $L_{\text{MI}}$, we aim to reduce the dependency between the gradients of different modality encoders, thereby reducing the risk of privacy leakage in MMFL.

# 5 Experiments

## 5.1 Datasets

We conduct MMFL experiments using both synthesized and native multimodal datasets. For image-text modality studies, we employ CIFAR-10/100 with text descriptions generated from image labels, following standard GIA experimental protocols. Additionally, we validate our method on authentic multimodal benchmarks: Hateful-Memes for social media content analysis and CrisisMMD for disaster response.

## 5.2 Evaluation Metrics

Following [46, 26, 29], we evaluate: (1) Privacy protection via Peak Signal-to-Noise Ratio (PSNR) between original/reconstructed images on CIFAR datasets, and LPIPS for complex images in CrisisMMD/Hateful-Memes; (2) Text Recovery Rate (TRR) measuring semantic similarity between original and recovered texts; (3) Except for employing AUC as the evaluation metric on the Hateful Memes dataset and F1-score on the CrisisMMD dataset, classification accuracy is utilized to measure model performance across all other datasets.

## 5.3 Attack Methods

We implement two gradient inversion attacks: (1) DLG [59] for CIFAR datasets with cross-modal label recovery, following the method in [29] and (2) IG [13] enhanced with stable diffusion generators for CrisisMMD/Hateful-Memes, where limited label semantics in these two datasets necessitate accelerated text-guided image reconstruction. Both methods optimize pseudo-inputs by minimizing the cosine distance between real and synthetic gradients through iterative backpropagation.

## 5.4 Baselines

Our method is compared against: DP-FedAvg [32] applying uniform noise across modalities; NbAFL [48] with client-side parameter noising; LDP-FL [40] using random response mechanisms; and DP-FedAvg-MI adjusting noise scales via mutual information analysis.

The model used in the experiment has two encoders and a classifier layer. On CIFAR-10 and CIFAR-100, the text encoder is based on LeNet-5 and the image encoder is based on TextCNN. On Crisis MMD and Hateful Memes, the text encoder is based on Bert and the image encoder is based on Resnet-50. For early fusion, the features generated by the encoders are fused by concatenation, and for late fusion, logits will be output after passing through the fully connected layer and fused by averaging at the decision level.

Table 1: Conditional mutual information comparison.

| Fusion Method | CIFAR-10 | CIFAR-100 |
|---|---|---|
| Text Early Fusion | 0.2539 | 0.2439 |
| Text Late Fusion | 0.5383 | 0.3909 |
| Image Early Fusion | 0.0100 | 0.0120 |
| Image Late Fusion | 0.0180 | 0.0390 |

We take $\lambda$ as 1e-2 chosen via grid search, $\delta$ as 1e-5 and clipping norm as 1.0. The batch size during training is set to 128. The learning rate $\eta$ is set to 1e-3 and the training is conducted for 200 rounds. We use neural estimators to calculate mutual information. We run all experiments on Intel Xeon Gold 6133 CPU, RTX4090 GPU.

## 5.5 Conditional mutual information can help measure leakage risk differences

We compare the text and image conditional mutual information between early fusion and late fusion MMFL on different datasets. It can be seen from Table 1 that for the model we used on the two data sets, the conditional mutual information of text modality is larger than that of the image modality, this trend is consistent with our observation in the actual attack experiment that the speed of text recovery is much faster than that of image recovery, which proves that conditional mutual information can be successfully applied to measure the difference of information leakage risk among different modes. Moreover, the conditional mutual information of each modality of the late fusion is higher than that of the early fusion. This may be because each encoder of the late fusion model is more independent than that of the early fusion model, so it has a greater impact on the overall gradient of the model

Table 2: Defense performance on CIFAR-10 and CIFAR-100.

| CIFAR-10 | $\epsilon = 0.25$ | | | $\epsilon = 0.5$ | | | $\epsilon = 1$ | | | $\epsilon = 2$ | | |
|---|---|---|---|---|---|---|---|---|---|---|---|---|
| Method | Acc↑ | PSNR↓ | TRR↓ | Acc↑ | PSNR↓ | TRR↓ | Acc↑ | PSNR↓ | TRR↓ | Acc↑ | PSNR↓ | TRR↓ |
| DP-FedAvg | 0.5219 | 6.563 | 0.57 | 0.6957 | 7.845 | 0.59 | 0.7336 | 8.166 | 0.61 | 0.8275 | 10.836 | 0.83 |
| NbAFL | 0.5654 | 7.039 | 0.61 | 0.7845 | 7.134 | 0.62 | 0.8031 | 8.998 | 0.72 | 0.8724 | 11.794 | 0.88 |
| LDP-FL | 0.1992 | 4.341 | 0.41 | 0.2173 | 6.931 | 0.51 | 0.4213 | 6.491 | 0.53 | 0.9037 | 7.713 | **0.68** |
| DP-FedAvg-MI | 0.5013 | 6.148 | 0.50 | 0.6724 | 7.073 | 0.52 | 0.7148 | 7.394 | 0.54 | 0.7987 | 9.757 | 0.76 |
| Sec-MMFL | **0.7875** | **4.192** | **0.39** | **0.8159** | **4.824** | **0.42** | **0.8624** | **5.142** | **0.51** | **0.9201** | 6.912 | 0.69 |

| CIFAR-100 | $\epsilon = 1$ | | | $\epsilon = 2$ | | | $\epsilon = 5$ | | | $\epsilon = 10$ | | |
|---|---|---|---|---|---|---|---|---|---|---|---|---|
| Method | Acc↑ | PSNR↓ | TRR↓ | Acc↑ | PSNR↓ | TRR↓ | Acc↑ | PSNR↓ | TRR↓ | Acc↑ | PSNR↓ | TRR↓ |
| DP-FedAvg | 0.3811 | 8.372 | 0.51 | 0.5233 | 12.592 | 0.74 | 0.6783 | 14.674 | 0.82 | 0.7538 | 17.515 | 0.91 |
| NbAFL | 0.4467 | 8.699 | 0.53 | 0.6185 | 11.482 | 0.71 | 0.7225 | 14.645 | 0.78 | 0.8321 | 18.014 | 0.89 |
| LDP-FL | 0.1089 | **6.255** | 0.45 | 0.4451 | 9.265 | 0.63 | 0.5154 | 11.197 | 0.66 | 0.5218 | 14.226 | 0.71 |
| DP-FedAvg-MI | 0.3704 | 7.628 | 0.49 | 0.4921 | 10.243 | 0.66 | 0.6561 | 13.588 | 0.79 | 0.7497 | 15.691 | 0.78 |
| Sec-MMFL | **0.4578** | 6.387 | **0.43** | **0.6759** | 9.241 | 0.59 | **0.8028** | 10.136 | 0.64 | **0.9214** | 12.891 | **0.67** |

and leaks more information, thus the barrel effect will be even stronger. Since it can be seen that the risk of privacy disclosure of all modalities of late fusion MMFL may be greater, we mainly use late fusion MMFL for privacy disclosure risk analysis in the following experiments.

## 5.6 Conditional mutual information can provide better protection

To measure the capability of Sec-MMFL to provide more balanced and reasonable privacy protection in MMFL, we compare the accuracy , similarity of the recovered text and similarity of the recovered image compared to the original training data. From Table 2 it can be seen that existing methods exhibit fragile protection capabilities because they fail to account for the differing information carried by different modalities and varying recovery rates. For instance,

| Method | Hateful Memes | | | Crisis MMD | | |
|---|---|---|---|---|---|---|
| | AUC↑ | LPIPS↑ | TRR↓ | F1↑ | LPIPS↑ | TRR↓ |
| DP-FedAvg | 0.433 | 0.766 | 0.62 | 0.338 | 0.536 | 0.55 |
| NbAFL | **0.617** | 0.573 | 0.77 | 0.384 | 0.572 | 0.58 |
| LDP-FL | 0.376 | 0.934 | 0.51 | 0.263 | 0.583 | 0.51 |
| DP-FedAvg-MI | 0.417 | 0.871 | 0.59 | 0.321 | 0.627 | 0.45 |
| Sec-MMFL | 0.564 | **0.975** | **0.47** | **0.385** | **0.801** | **0.39** |

Table 3: Comparison of defense performance.

higher text recovery ability can leak more information to the image modality, leading to higher image similarity — a sign of insufficient privacy protection. The method of using simple mutual information between the data of each modality and the gradient(DP-FedAvg-MI) can make the protection ability of each mode of MMFL more balanced. However, it does not take into account the relationship between the input of other modalities and the gradient, it sometimes cannot correctly reflect the real leakage risk of each modality, resulting in the protection effect is not balanced enough. Sec-MMFL using conditional mutual information can reasonably measure the risk of privacy breach, make the protection effect of the two modes more balanced, and achieve the minimum loss of accuracy simultaneously. In fact, not only when using LDP in MMFL, this method of using conditional mutual information to leakage risk assessing and noise adjusting can also be applied on other defense methods in MMFL such as directly adding Gaussian noise on source data, which can also help to achieve a more balanced distribution of noise and more reasonable protection. Same trend is shown in Table 3 and Figure 4.

## 5.7 Task performance loss of Sec-MMFL

In order to explore the task performance cost for more balanced MMFL privacy protection under different settings, we conducted experiments with different client number, privacy budget amount, and different client data heterogeneity distribution. As shown in Figure 5a and 5b that as the number of clients goes up from 5 to 30, the performance loss has not changed much, but when the privacy budget $\epsilon$ tightens from 1 to 0.5, the performance loss increases slightly, we think it may be because smaller privacy budgets will bring more noise and let MMFL model be more sensitive to noise reallocation. As for the performance loss when the data distribution heterogeneity of each client changes, we conducted experiments on the two data sets at the setting of $\epsilon = 1$ on CIFAR-10 and

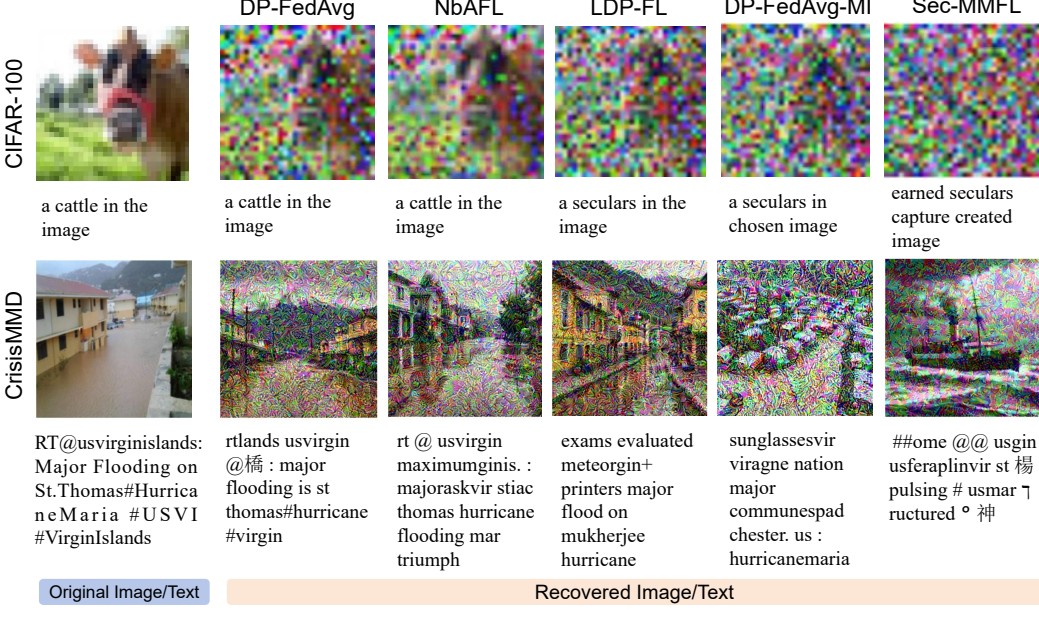

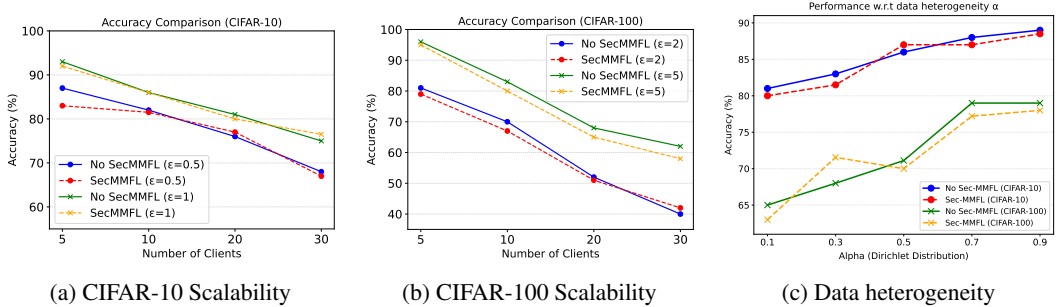

Figure 4: Attack effect visualization.

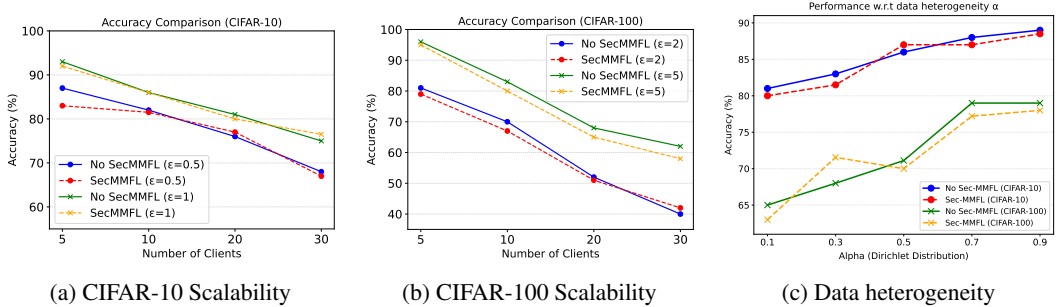

(a) CIFAR-10 Scalability     (b) CIFAR-100 Scalability     (c) Data heterogeneity

Figure 5: Parameter Sensitivity Analysis.

$\epsilon = 5$ on CIFAR-100 for 10 clients. It can be seen from Figure 5c that when data heterogeneity is low (i.e., larger Dirichlet parameter $\alpha$, such as 0.9 or 0.7), Sec-MMFL has relatively small impact on performance; as heterogeneity increases (smaller $\alpha$, down to 0.1), accuracy remains stable on both datasets, with slightly larger fluctuations on CIFAR-100 likely due to its more complex task.

## 6 Conclusion and Future Work

We propose Sec-MMFL, a framework that help enhance privacy preservation in MMFL. By leveraging information theory, Sec-MMFL can reasonably assess the privacy leakage risk of each modality in MMFL and properly adjust the privacy protection intensity of each modality, making MMFL more robust against GIA attacks. Extensive experiments conducted on various settings show that Sec-MMFL strikes the best balance between defensive effect and utility. Although our method has shown significantly more balanced and robust protection in MMFL of image and text modalities, there is currently no work to perform better privacy protection studies on more modalities, and future work can be carried out on more baseline models and other modalities.

**Acknowledgment** This work is supported by the National Key Research and Development Program of China under grant 2024YFC3307900; the National Natural Science Foundation of China under grants 62302184, 62376103, 62436003 and 62206102; Major Science and Technology Project of Hubei Province under grant 2024BAA008; Hubei Science and Technology Talent Service Project under grant 2024DJC078; Ant Group through CCF-Ant Research Fund; and Fundamental Research Funds for the Central Universities under grant YCJJ20252319. The computation is completed in the HPC Platform of Huazhong University of Science and Technology.

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
