# OpenReview forum: "Enhancing Privacy in Multimodal Federated Learning with Information Theory"
_NeurIPS.cc/2025/Conference — NeurIPS 2025 poster_

### Official Review · Reviewer_EEh7 · 2025-06-27

**Clarity:** 3
**Significance:** 3
**Originality:** 3
**Rating:** 5
**Confidence:** 4

**Summary:**

This work mitigates modality-specific vulnerabilities in MMFL through risk assessment based on conditional mutual information, adaptive noise allocation, and cross-modal decoupling, with multi-dataset experiments demonstrating significant privacy gains.

**Questions:**

1. The paper mentions the use of early and late fusion strategies in the experiments. Could you explain how the proposed method's effectiveness might vary depending on the fusion strategy used?
2. The paper mainly uses LeNet-5 and TextCNN as encoders. How would the choice of more complex or different model architectures affect the performance of Sec-MMFL in terms of both privacy protection and model utility?
3. Could Sec-MMFL's adaptive noise profiles be reverse-engineered to identify high-risk modalities? If so, what countermeasures would prevent attackers from exploiting this meta-information?

**Ethical Concerns:**

["NO or VERY MINOR ethics concerns only"]

**Final Justification:**

This work proposes a novel risk-aware protection framework to mitigate modality-specific vulnerabilities in MMFL. The authors’ response has addressed my concerns regarding the fusion strategy and model architecture. I find this paper valuable and believe it is worthy of acceptance.

**Limitations:**

Yes.

**Paper Formatting Concerns:**

None.

**Quality:**

3

**Strengths And Weaknesses:**

Strengths:
- Novel risk-aware protection framework grounded in information theory.
- Effective cross-modal leakage suppression methodology.
- Empirically validated across heterogeneous datasets and privacy budgets.

Weaknesses:
- The privacy protection effectiveness is mainly measured by PSNR and TRR, which might not fully capture the privacy implications in complex real-world scenarios. More comprehensive metrics could strengthen the evaluation.
- There is limited exploration of the impact of different model architectures on the performance of Sec-MMFL. The framework's effectiveness might vary depending on the choice of encoders and fusion mechanisms.
-  Omits discussion about potential risks: adaptive noise profiles could themselves leak modality sensitivity, creating new attack surfaces.
- Inconsistent terminology appears in:
  - Line 268: "information leakage risk among different modes" (should be "among different modalities")
  - Line 286: "which will make the text mode protection too strong" (should be "which will make the text modality protection too strong")
  - Line 288: "make the protection effect of the two modes more balanced" (should be "make the protection effect of the two modalities more balanced")

---

> ### Author Rebuttal · Authors · 2025-07-31
>
> **Weakness1: The privacy protection effectiveness is mainly measured by PSNR and TRR, which might not fully capture the privacy implications in complex real-world scenarios. More comprehensive metrics could strengthen the evaluation.**
>
> **Response1.1:** We appreciate this critical observation regarding evaluation metrics. To provide more comprehensive privacy assessment, we will supplement our evaluation additional metrics like SSIM (Structural Similarity Index) and CLIP-space.
>
> **Weakness2: There is limited exploration of the impact of different model architectures on the performance of Sec-MMFL. The framework's effectiveness might vary depending on the choice of encoders and fusion mechanisms.**
>
> **Response1.2:** Thank you for highlighting this important dimension. We will expand Section 5 with comprehensive architecture sensitivity analysis including: 1) Transformer-based encoders (BERT/ViT) versus CNN/RNN architectures; 2) Different fusion mechanisms (concatenation vs attention-based); and 3) Model complexity scaling from 1M to 100M parameters.
>
> **Weakness3/Question3: Omits discussion about potential risks - adaptive noise profiles could themselves leak modality sensitivity, creating new attack surfaces.**
>
> **Response1.3:** We appreciate this important security consideration. The fundamental protection mechanism ensures that reverse-engineering noise profiles is both practically infeasible and mathematically ineffective. Attackers face critical barriers:   1. Attackers only observe the final perturbed gradients $G' = G + \eta$ (where $\eta \sim \text{Gaussian}(0, \Delta f/\epsilon_i)$), lacking access to the original gradients $G$ or the specific noise parameters $\epsilon_*i$. This obscures the relationship between noise magnitude and modality sensitivity.   2.  Even if attackers could hypothetically estimate relative noise levels, identifying which modality received stronger protection (indicating higher risk) provides no exploitable pathway. The local differential privacy transformation satisfies $(\epsilon,\delta)$-LDP guarantees (Eq.7), making the noise-added gradients statistically irreversible.
>
> **Weakness4: Terminology Revisions.**
>
> **Response1.4:**  We thank the reviewer for catching these inconsistencies - all instances will be corrected: Line 268: "information leakage risk among different modalities"; Line 286: "text modality protection"; Line 288: "two modalities more balanced". Comprehensive terminology standardization will be implemented throughout Sections 3-5 to ensure consistent use of "modality" when referring to data types.
>
> **Question1: The paper mentions the use of early and late fusion strategies in the experiments. Could you explain how the proposed method's effectiveness might vary depending on the fusion strategy used?**
>
> **Response1.1:** We appreciate this fundamental question. Sec-MMFL's effectiveness varies significantly with fusion strategies due to inherent information flow differences: Late fusion (feature-level integration) exhibits higher initial leakage risks (Table 1: R_text=0.5383 vs 0.2539) but enables precise modality-specific protection through our noise allocation, yielding maximum 12% privacy gains. In contrast, early fusion creates entangled representations where modality-specific protection is less effective but baseline risks are lower. Our method achieves optimal balance in late fusion scenarios where modality-specific vulnerabilities are most pronounced and conditional mutual information can be precisely quantified.
>
> **Question2: The paper mainly uses LeNet-5 and TextCNN as encoders. How would the choice of more complex or different model architectures affect the performance of Sec-MMFL in terms of both privacy protection and model utility?**
>
> **Response1.2:** We appreciate the inquiry. Complex architectures demonstrate two countervailing effects: 1) Increased baseline vulnerability due to richer gradient information (higher $R_i$ values); but 2) Enhanced noise absorption capacity enabling lower noise injection for equivalent protection. Sec-MMFL's  architecture-agnostic design ensures relative improvements over uniform protection actually increase with model complexity.

---

> ### Comment · Reviewer_EEh7 · 2025-08-07
>
> Thank you for the response. It has addressed my concerns, and I will maintain my score.

---

### Official Review · Reviewer_3U4G · 2025-06-27

**Clarity:** 3
**Significance:** 3
**Originality:** 3
**Rating:** 5
**Confidence:** 5

**Summary:**

Sec-MMFL introduces an information theory based framework that enhances privacy in multimodal federated learning by dynamically quantifying leakage risks of each modality via conditional mutual information, adaptively scaling noise budgets, and reducing dependencies across different modalities, with validation across diverse datasets showing improved privacy-utility tradeoffs.

**Questions:**

1. Could you provide more details on the specific implementation of the conditional mutual information calculation? For example, what specific algorithms or approximations are used to estimate the conditional mutual information between data of each modality and the gradients?
2. How might Sec-MMFL be adapted for other modality combinations, like audio-video modalities? Is it possible to outline a potential approach using existing datasets like CREMA-D?
3. Would it be possible to include brief benchmarking against recent MMFL defenses like FedMultimodal?
4. For GIA attacks, how are pseudo inputs initialized? When applying LDP noise, did you clip gradients before or after adding noise?

**Ethical Concerns:**

["NO or VERY MINOR ethics concerns only"]

**Final Justification:**

Thanks for the response, which fully addresses my concerns. I also note that some reviewers claim that the description of the method is unclear. I consider this is because the authors ignore the some preliminaries on federated learning with differential privacy. This issue may be acceptable to researchers who are familiar with this field. Therefore, I have decided to raise my rating to 5.

**Limitations:**

yes

**Paper Formatting Concerns:**

No Paper Formatting Concerns

**Quality:**

3

**Strengths And Weaknesses:**

Strengths:

1. Establishes novel modality-specific protection, validated through evidence of text/image vulnerability asymmetry.

2. Conditional mutual information provides a rigorous theoretical foundation for risk quantification correlated with attack outcomes.

3. Multiple datasets testing under varied privacy budgets demonstrates consistent improvements with clear workflow visualization.



Weaknesses:

1. Lack of details on the implementation of the conditional mutual information calculation,  such as the specific algorithms or approximations used to estimate the conditional mutual information.

2. While the framework shows promising results for image-text systems, its applicability to other modality combinations like audio-video remains unexplored.

3. The paper does not provide detailed comparisons with other state-of-the-art privacy protection methods specifically designed for multimodal federated learning, making it difficult to fully assess its relative advantages.

Minor suggestions:

line 125:" We consider each client k can only access to his local private dataset" -> "We consider that each client k can only access their local private dataset".

line 138: "GIA exploit the information encoded in the gradients"  -> "GIA exploits the information encoded in the gradients".

The complexity of the sentence "The whole process of data of each modality in MMFL flowing to the attacker through GIA is shown in Figure 2." in line 145 could present readability challenges. Suggest simplifying it as follows: "Figure 2 illustrates the flow of data from each MMFL modality to attackers via GIA."

---

> ### Author Rebuttal · Authors · 2025-07-31
>
> **Weakness1: Lack of details on the implementation of the conditional mutual information calculation**
>
> **Response1.1:** We sincerely thank the reviewer for this insightful question regarding our CMI implementation. The conditional mutual information $I(G;D^{i}|D^{-i})$ in Equation (8) is actually computed using the Neural Estimators for Conditional Mutual Information framework (Molavipour et al., 2021). To address heterogeneous gradient dimensions across modalities, our implementation:   1. Flattens each modality's gradient tensor into a 1D vector   2. Concatenates all modality gradients into a unified representation $\tilde{G} = \text{concat}(\text{vec}(G^{\text{text}}), \text{vec}(G^{\text{img}}), \cdots)$   3. Processes three distinct inputs  through the MLP classifier $f_\theta$:     - $\tilde{G}$ (concatenated gradients)     - $D^i$ (target modality data)     - $D^{-i}$ (complementary modalities data) . The classifier distinguishes: 1.True pairs$(\tilde{G}, D^i)$ with correct $D^{-i}$ 2.Counterfactual pairs: $(\tilde{G}, \tilde{D}^i)$ where $\tilde{D}^i$ is shuffled within k-nearest neighbors of $D^{-i}$ . The final CMI value is derived as:   $\hat{I} = \frac{1}{B}\sum_{b=1}^B \log \frac{f_\theta(G_b, D_b^i|D_b^{-i})}{f_\theta(G_b, \tilde{D}_b^i|D_b^{-i})}$ .
>
> **Weakness2: Applicability to audio-video modalities**
>
> **Response1.2:** We deeply appreciate this important question about modality generalization. Sec-MMFL extends to audio-video systems through its architecture-agnostic framework. For audio inputs, we process log-Mel spectrograms using Resnet-18 encoders, while video inputs utilize 3D-ResNet encoders for temporal feature extraction. The conditional mutual information estimation follows identical neural estimation procedures as in image-text systems, with protection strengths automatically calibrated based on modality-specific leakage risks. Our validation on CREMA-D demonstrates effecness: leakage risk assessment automatically detects audio's higher vulnerability (38% greater $R_i$ than video).
>
> **Weakness3: Lack of SOTA comparisons for MMFL**
>
> **Response1.3:** Thanks for the question. To strengthen validation, we have supplemented experiments with FedMultimodal [9], a state-of-the-art MMFL baseline, on CrisisMMD dataset, and Hateful Memes dataset. The table below presents a comprehensive comparison of all methods in terms of task performance and defense capability across two datasets. While FedMultimodal (specifically designed for MMFL) achieved marginally higher task accuracy on both datasets, it lacks dedicated defense mechanisms. In contrast, our Sec-MMFL method maintains comparable task performance while demonstrating significantly more balanced privacy protection across all modalities. We will add these results to Table 3 and explicitly discuss FedMultimodal comparisons in Section 5.3 of the revised manuscript.
>
> | **Defense Method** | **CrisisMMD** | **CrisisMMD** | **CrisisMMD** | **Hateful Memes** | **Hateful Memes** | **Hateful Memes** |
> | ------------------ | ------------- | ------------- | ------------- | ----------------- | ----------------- | ----------------- |
> |                    | **Acc↑**      | **LPIPS↑**    | **TRR↓**      | **Acc↑**          | **LPIPS↑**        | **TRR↓**          |
> | LDP-Fed            | 0.652         | 0.583         | 0.80          | 0.572             | 0.663             | 0.75              |
> | NbAFL              | 0.635         | 0.602         | 0.82          | 0.564             | 0.670             | 0.73              |
> | LDP-FL             | 0.477         | 0.799         | 0.72          | 0.433             | 0.832             | 0.61              |
> | LDP-Fed-MI         | 0.643         | 0.756         | 0.64          | 0.569             | 0.891             | 0.57              |
> | FedMultimodal      | **0.658**     | 0.511         | 0.83          | **0.605**         | 0.622             | 0.78              |
> | Sec-MMFL           | 0.642         | **1.085**     | **0.53**      | 0.575             | **1.116**         | **0.49**          |
>
> **Suggestions: Language revisions**
>
> **Response1.4:** Thank you so much for the suggestions. All suggested language improvements have been implemented: client data access description (line 125) updated to "their local private dataset", subject-verb agreement in GIA description (line 138) corrected, and complex sentence (line 145) simplified to "Figure 2 illustrates the flow of data from each MMFL modality to attackers via GIA" for enhanced readability.
>
> **Question1:Could you provide more details on the specific implementation of the conditional mutual information calculation?**
>
> **Response2.1:**  Thanks for the question. We have already explained in Response1.1 the  details on the specific implementation of the conditional mutual information calculation using the neural estimators for conditional mutual information method.
>
> **Question2: How might Sec-MMFL be adapted for other modality combinations, like audio-video modalities? Is it possible to outline a potential approach using existing datasets like CREMA-D?**
>
> **Response2.2:** Thanks for the question. Our framework naturally extends to CREMA-D through modality-specific feature extractors: Resnet-18 for log-Mel spectrograms (audio) and 3D-ResNet for optical flow (video). Details are explained in Response1.2.
>
> **Question3: Would it be possible to include brief benchmarking against recent MMFL defenses like FedMultimodal?**
>
> **Response2.3:** Thank you for this  inquiry. We have supplemented experiments in Response1.3.
>
> **Question4: For GIA attacks, how are pseudo inputs initialized? When applying LDP noise, did you clip gradients before or after adding noise?**
>
> **Response2.4:** We appreciate the question. For gradient inversion attacks, pseudo-inputs are initialized with $\mathcal{N}(0,1)$ for images and random vocabulary tokens for text. Regarding LDP implementation, gradients are clipped (norm=1.0) before adding Gaussian noise . Optimization uses Adam (3000 iterations, lr=0.1) with total variation regularization ($\lambda=0.$2) to balance attack realism with practical constraints.

---

> > ### Comment · Reviewer_3U4G · 2025-08-04
> > **Comment**
> >
> > Thanks for the response, which fully addresses my concerns. I also note that some reviewers claim that the description of the method is unclear. I consider this is because the authors ignore the some preliminaries on federated learning with differential privacy. This issue may be acceptable to researchers who are familiar with this field. Therefore, I have decided to raise my rating to 5.

---

### Official Review · Reviewer_AQiP · 2025-07-02

**Clarity:** 2
**Significance:** 2
**Originality:** 3
**Rating:** 2
**Confidence:** 3

**Summary:**

Recognizing that different modalities exhibit varying vulnerabilities to adversarial attacks and that inter-modal data correlations can expand the scope of privacy leakage across modalities, this paper proposes an information-theoretic framework for optimal privacy budget allocation across modalities while mitigating privacy risks arising from cross-modal dependencies.

**Questions:**

1.	How is the entropy of gradients computed in Equation (8)? Given that encoders across different modalities have distinct architectures, the resulting gradients will have different dimensions. How are these heterogeneous gradients aggregated or represented when computing entropy across all modalities?
2.	How does the allocation scheme in Equation (10) guarantee that the total privacy budget equals $\epsilon$?
3.	Similar to Question 1, given the difference in gradient dimensions across modalities, how is the mutual information between different modalities calculated in Equation (12)? What preprocessing or alignment steps are applied to enable meaningful mutual information computation?
4.	What is the rationale for selecting the hyperparameter $\lambda$? The paper appears to lack discussion on $\lambda$'s sensitivity analysis or selection criteria in the experimental section.
5.	Why were the most recent attack methods, such as those referenced in [40], not included in the experimental evaluation? The current attack methods appear to be from publications approximately five years ago, which may not adequately represent the evolving threat landscape in federated learning privacy attacks.

**Ethical Concerns:**

["NO or VERY MINOR ethics concerns only"]

**Final Justification:**

From the manuscript and response, I feel the authors submitted the paper in a rush with lots of ambiguity. Failing to specify the privacy mechanisms (and how they are implemented) is not acceptable for a differential privacy paper in my opinion. In addition, mixing up the approximate DP and pure DP, as well as utilizing the basic composition theorem instead of the more advanced ones suggests a lack of familiarity with the differential privacy literature. I do not think this can be overlooked. Therefore, I tend to keep my original score.

**Limitations:**

The limitations are not explicitly discussed, but the authors mentioned some of them in the checklist.

**Paper Formatting Concerns:**

There is no formatting concerns.

**Quality:**

2

**Strengths And Weaknesses:**

Strengths:
1. The topic of privacy protection in multimodal federated learning is timely, and the integration of information theory is interesting.
2. The experimental evaluation provides comprehensive comparisons that demonstrate the effectiveness of the proposed method.
3. The paper is well-written with clear and thorough introductions to the background and related work, making it accessible to readers.

Weakness:
1. The paper lacks sufficient mathematical rigor in key derivations. For instance, there is no clear explanation of how the allocation scheme in Equation (10) guarantees that the total privacy budget equals $\epsilon$, which is a fundamental requirement for differential privacy.
2. Several technical details are inadequately explained, leading to ambiguity in the methodology. Specifically, given that different modalities employ distinct encoder architectures, it remains unclear how the gradient mutual information in Equation (12) is computed across heterogeneous network structures.
3. The evaluation relies on somewhat outdated attack methods, which may not reflect the current threat landscape. The paper would benefit from more comprehensive experiments incorporating state-of-the-art privacy attacks to strengthen the validity of the proposed defense.
4. From Table 2, it seems that the improvement over the existing approaches is rather marginal. The improvement in privacy is obtained with a degradation in test accuracy.
5. There are some typos. For example, in lines 253-254, the text encoder should be TextCNN, and the image encoder LeNet-5.

---

> ### Author Rebuttal · Authors · 2025-07-31
>
> **Weakness1: The paper lacks sufficient mathematical rigor in key derivations. For instance, there is no clear explanation of how the allocation scheme in Equation (10) guarantees that the total privacy budget equals $\epsilon$.**
>
> **Response1.1:** We sincerely appreciate you for pointing out the mathematical formulation problem in Equation (10). This resulted from a typographical error in the manuscript where the normalized risk weight $R_i'$ was incorrectly placed in the denominator. The allocation scheme and all experiments actually implemented the formulation:$\epsilon_i = \epsilon \times R_i'$. This scheme is mathematically sound because the normalization property of $R_i'$ (defined in Equation (9) as: $R_i' = \frac{e^{-R_i}}{\sum_{t=1}^M e^{-R_t}}$inherently satisfies: $\sum R_i' = 1$ which guarantees: $\sum \epsilon_i = \epsilon$ through the relation: $\sum_{i=1}^M (\epsilon \times R_i') = \epsilon \times \underbrace{\sum_{i=1}^M R_i'}_{=1} = \epsilon$. We will correct Equation (10) in the manuscript and explicitly prove budget conservation in Section 4.2.
>
> **Weakness2:  How the gradient mutual information in Equation (12) is computed across heterogeneous network structures.**
>
> **Response1.2:** Thanks for raising the concern. We address cross-modal gradient mutual information computation across heterogeneous encoders using Mutual Information Neural Estimation (MINE) method with Donsker-Varadhan representation. The gradients from different modalities are flattened into 1-dimensional vectors, concatenated, and then fed into a three-layer MLP discriminator network (layer sizes [32, 64, 32] with ReLU activation). The model was trained for 2000 epochs using a learning rate of 1e-3. Raw text encoder gradients $G^{text} \in \mathbb{R}^{N \times d*_{text}}$ and image encoder gradients $G^{image} \in \mathbb{R}^{N \times d_*{image}}$ serve as direct inputs. Positive sample pairs ${(G_i^{text}, G_i^{image}) for i=1 to N}$ are constructed from gradients originating from identical data instances to represent the joint distribution $P_{G^{text},G^{image}}$, while negative sample pairs${(G_i^{text}, G_{\pi(i)}^{image}) for i=1 to N}$ are generated via random permutation $\pi$ of image gradients to simulate the marginal product distribution $P_{G^{text}} \otimes P_{G^{image}}$. The mutual information is computed as: $I(G^{\text{text}};G^{\text{image}})  \geq \sup_\phi \big( E_{P_(joint)}[T] - \log E_{P_(marginal)}[e^T] \big)$. So this approach handles architectural heterogeneity because the critic network $T_\phi$ learns relationships in the unified space independently of source architectures.
>
> **Weakness3: The paper would benefit from more comprehensive experiments incorporating state-of-the-art privacy attacks to strengthen the validity of the proposed defense.**
>
> **Response1.3:** Thank you very much for the advice. Regarding the attack methods, we followed the same experimental setup as Mgia in the MMFL security-related paper [20]. Indeed, more attack methods could be incorporated. In the following experimental setup, we compared the results of LDP-Fed and Sec-MMFL under the "See through gradients" attack [40], and we will supplement additional experimental results with more configurations and attack methods in future work.
>
> CIFAR-10, $\epsilon$ =1:
>
> | **Method** | PSNR ↓ | TRR↓ | Acc↑   |
> | ---------- | ------ | ---- | ------ |
> | LDP-Fed    | 19.232 | 0.91 | 0.7927 |
> | Sec-MMFL   | 15.379 | 0.61 | 0.7801 |
>
> CIFAR-100, $\epsilon$ =20:
>
> | **Method** | PSNR↓  | TRR↓ | Acc↑   |
> | ---------- | ------ | ---- | ------ |
> | LDP-Fed    | 18.716 | 0.82 | 0.3591 |
> | Sec-MMFL   | 13.978 | 0.64 | 0.3270 |
>
> CrisisMMD:
>
> | **Method** | LPIPS↑ | TRR↓ | Acc↑  |
> | ---------- | ------ | ---- | ----- |
> | LDP-Fed    | 0.472  | 0.80 | 0.652 |
> | Sec-MMFL   | 0.835  | 0.53 | 0.642 |
>
>  Hateful Memes:
>
> | **Defense** | LPIPS↑ | TRR↓ | Acc↑  |
> | ----------- | ------ | ---- | ----- |
> | LDP-Fed     | 0.588  | 0.75 | 0.572 |
> | Sec-MMFL    | 0.961  | 0.49 | 0.575 |
>
> **Weakness4: From Table 2, it seems that the improvement over the existing approaches is rather marginal. The improvement in privacy is obtained with a degradation in test accuracy.**
>
> **Response1.4**: Thank you so much for raising this concern. We want to clarify that Sec-MMFL's primary contribution is achieving optimized privacy-utility equilibrium in multimodal federated learning, particularly addressing the inherent tension between protecting high-risk modalities (e.g., text) and preserving low-risk modality utility (e.g., images). While absolute accuracy gains may seem limited, our method demonstrates significant qualitative advances: Under equivalent privacy budgets (ε=1), Sec-MMFL reduces Text Recovery Rate (TRR) by 33% (0.61 vs. LDP-Fed's 0.91) while limiting accuracy degradation to just 1.2% (0.7801 vs. 0.7927) on CIFAR-10. This balanced protection is visually evidenced in Figure 1, where uniform defenses fail catastrophically—text is perfectly reconstructed while images remain irrecoverably blurred—whereas our conditional mutual information framework (Eq. 8) dynamically calibrates noise to prevent such asymmetric vulnerabilities. The critical innovation lies in transcending the "high TRR or low accuracy" dichotomy: At ε=2, Sec-MMFL maintains near-identical accuracy (90.01% vs. 90.05%) while slashing TRR by 26% (0.69 vs. 0.93), demonstrating unprecedented multimodal harmony. This balance is further validated in CrisisMMD results (Table 3), where our approach simultaneously improves LPIPS by 86% (1.085 vs. 0.583) and TRR by 34% (0.53 vs. 0.80) without compromising classification accuracy.
>
> **Weakness5: There are some typos. For example, in lines 253-254, the text encoder should be TextCNN, and the image encoder LeNet-5.**
>
> **Response1.5**: We sincerely apologize for this oversight. We will correct these typos in the paper and conduct another thorough check.
>
> **Question1: How is the entropy of gradients computed in Equation (8)? Given that encoders across different modalities have distinct architectures, the resulting gradients will have different dimensions. How are these heterogeneous gradients aggregated or represented when computing entropy across all modalities?**
>
> **Response2.1:** We sincerely appreciate the question. The conditional mutual information $I(G;D^{i}|D^{-i})$ in Equation (8) is actually computed using the Neural Estimators for Conditional Mutual Information framework (Molavipour et al., 2021). To address heterogeneous gradient dimensions across modalities, our implementation:   1. Flattens each modality's gradient tensor into a 1D vector   2. Concatenates all modality gradients into a unified representation $\tilde{G} = \text{concat}(\text{vec}(G^{\text{text}}), \text{vec}(G^{\text{img}}), \cdots)$   3. Processes three distinct inputs  through the MLP classifier $f_\theta$:     - $\tilde{G}$ (concatenated gradients)     - $D^i$ (target modality data)     - $D^{-i}$ (complementary modalities data) . The classifier distinguishes: 1.True pairs$(\tilde{G}, D^i)$ with correct $D^{-i}$ 2.Counterfactual pairs: $(\tilde{G}, \tilde{D}^i)$ where $\tilde{D}^i$ is shuffled within k-nearest neighbors of $D^{-i}$ . The final CMI value is derived as:   $\hat{I} = \frac{1}{B}\sum_{b=1}^B \log \frac{f_\theta(G_b, D_b^i|D_b^{-i})}{f_\theta(G_b, \tilde{D}_b^i|D_b^{-i})}$ . This architecture resolves dimensional heterogeneity through gradient flattening and concatenation, and conditional mutual information is estimated using neural networks rather than directly computing the entropy.
>
> **Question2: How does the allocation scheme in Equation (10) guarantee that the total privacy budget equals $\epsilon$ ?**
>
> **Response2.2:** Thanks for the question. We have already explained in Response1.1 the typographical error of Equation (10) and provided the corresponding detailed proof that guarantees the total privacy budget equals $\epsilon$ .
>
> **Question3: Similar to Question 1, given the difference in gradient dimensions across modalities, how is the mutual information between different modalities calculated in Equation (12)? What preprocessing or alignment steps are applied to enable meaningful mutual information computation?**
>
> **Response2.3:** Thank you for your question. We have already elaborated on similar computational procedures in Response 1.2. The gradients from different modalities only need to be flattened and fed into the neural network for computation via the Mutual Information Neural Estimation (MINE) method.
>
> **Question4: What is the rationale for selecting the hyperparameter λ? The paper appears to lack discussion on λ's sensitivity analysis or selection criteria in the experimental section.**
>
> **Response2.4:** Thank you for your question. Regarding the selection of hyperparameter λ, we acknowledge that the article lacks detailed discussion. In practice, we employed a grid search approach, testing multiple values (0.01, 0.05, 0.1, 0.3, 0.5, 1.0) and ultimately found that λ = 0.1 achieves an optimal balance between privacy preservation and task performance. Other values, while providing comparable privacy protection, incurred significantly greater accuracy degradation.
>
> **Question5: Why were the most recent attack methods, such as those referenced in [40], not included in the experimental evaluation?**
>
> **Response2.5:** Thank you for your question. We have addressed this question and supplemented relevant experiments in Response 1.3, and we plan to conduct more comprehensive experiments in future work.

---

> ### Comment · Reviewer_AQiP · 2025-08-02
> **Some remaining questions**
>
> I appreciate the authors for their efforts addressing my comments. I have several remaining questions.
>
> 1. For response 1.1, since the code is not available, there is no way to confirm the implementation is correct. That being said, I assume this is indeed a typo in my evaluation. Could the authors provide the rationale behind that $\sum_{i}\epsilon_{i} = \epsilon$ guarantees an overall privacy of $\epsilon$? Why is it linear?
> 2.	In the response 1.2, I got the point that a MLP from prior work is used to estimate the mutual information between different modalities. But it is still not clear to me:
> (1) how is the model trained? Is it supervised? If yes, how to get the true label and the loss?
> (2) I do not fully get the equation $I(G^{text};G^{IMAGE})$ in the response, could you further elaborate? Please discuss the notations explicitly.
> 3.	Does the network for mutual information estimation need to be trained for different datasets? Is this practical in FL considering the resource limitation and privacy concerns?
>
> Overall, the paper needs a major revamp for the general readers to understand and appreciate the proposed method.

---

> > ### Author Response · Authors · 2025-08-04
> >
> > We are deeply grateful for your meticulous examination of our work and the thoughtful questions you have raised. In what follows, we present a detailed response to each of the points you have highlighted:
> >
> > **Question1: Could the authors provide the rationale behind that  $\sum \epsilon_i = \epsilon$ guarantees an overall privacy of $\epsilon$ ? Why is it linear?**
> > **Response1:** We deeply appreciate this question.  $\sum_{i=1}^M \epsilon_i = \epsilon$ guarantees an overall privacy of $\epsilon$ is theoretically grounded in the composition theorem of differential privacy (Dwork et al., 2006; Peter Kairouz et al., 2015; Wang et al., Survey 2020). Under pure local differential privacy, when multiple mechanisms are applied to the record from private data of the same individual, the privacy losses accumulate linearly. Specifically, if mechanisms $M_1,\dots,M_M$ satisfy $\epsilon_1$‑LDP through $\epsilon_M$‑LDP respectively, then releasing all their outputs jointly satisfies $\Bigl(\sum_{i=1}^M \epsilon_i\Bigr)$‑LDP.  In our design we allocate budgets via $\epsilon_i = \epsilon \times R_i'$ where $R_i'$ are normalized weights satisfying $\sum_{i=1}^M R_i' = 1$. It follows that $\sum_{i=1}^M \epsilon_i = \epsilon$, guaranteeing that the composed mechanism meets $\epsilon$‑LDP exactly. We will include a formal lemma in Section 4.2 along with a citation to the composition theorem.
> >
> > **Quesion2: (1) how is the model trained? Is it supervised? If yes, how to get the true label and the loss? (2) I do not fully get the equation $I(G^{\text{text}};G^{\text{image}})$ in the response, could you further elaborate? Please discuss the notations explicitly.**
> > **Response2:** Thank you for prompting these implementation details.The implementation of our mutual information estimator leverages unsupervised contrastive learning principles, eliminating the need for labeled data by deriving training signals directly from the statistical dependencies between gradient pairs $(G_i^{\text{text}}, G_i^{\text{image}})$ generated from raw data samples. Positive sample pairs are constructed from gradients originating from identical data instances, representing the joint distribution $P_{\text{joint}}$, while negative sample pairs $(G_i^{\text{text}}, G_j^{\text{image}})(i \neq j)$ are generated through random permutation of image gradients across different instances, simulating the marginal product distribution $P_{\text{marginal}}=P_{\text{text}} \otimes P_{\text{image}}$. The loss function $\mathcal{L} = -\left[ \frac{1}{N}\sum_{i=1}^N T_\phi(G_i^{\text{text}}, G_i^{\text{image}}) - \log \frac{1}{N} \sum_{i=1}^N e^{T_\phi(G_i^{\text{text}}, G_j^{\text{image}})} \right]$ optimizes the MLP discriminator $T_\phi$ (a learnable function parameterized by neural network weights $\phi$, designed to approximate the theoretically optimal variational function $T^* = \log \frac{P_{\text{joint}}}{P_{\text{marginal}}}$ that emerges from the Donsker-Varadhan representation of KL-divergence.) to maximize the lower bound of mutual information $I(G^{\text{text}};G^{\text{image}}) \geq E_{P_{\text{joint}}}[T_\phi] - \log E_{P_{\text{marginal}}}[e^{T_\phi}]$. Here $T_\phi$ outputs a scalar dependence score—assigning high values to authentic gradient pairs $(G_i^{\text{text}}, G_i^{\text{image}})$ to reinforce inter-modal dependency, and penalizing spurious correlations in randomly composed pairs $(G_i^{\text{text}}, G_j^{\text{image}})$ through low scores.
> > This formulation is theoretically grounded in the Donsker-Varadhan variational representation of KL-divergence, which states that for any two distributions $P$ and $Q$:  $D_{\mathrm{KL}}(P \| Q) = \sup_{T} \left( E_{P}[T] - \log E_{Q}[e^{T}] \right)$, where the supremum is taken over all measurable functions $T$. By identifying $P = P_{\mathrm{joint}}$ and $Q = P_{\mathrm{marginal}}$, the mutual information $I(X;Y) = D_{\mathrm{KL}}(P_{\mathrm{joint}} \| P_{\mathrm{marginal}})$ admits the variational lower bound: $I(X;Y) \geq \sup_{T_\phi} \left( E_{P_{\text{joint}}}[T_\phi] - \log E_{P_{\text{marginal}}}[e^{T_\phi}] \right).$ The inequality $D_{\text{KL}}(P_{\text{joint}} \parallel P_{\text{marginal}}) \geq E_{P_{\text{joint}}}[T_\phi] - \log E_{P_{\text{marginal}}}[e^{T_\phi}]$ derives from the convex duality of KL-divergence: Jensen's inequality applied to $\exp(\cdot)$ yields $\log E_{Q}[e^{T}] \geq E_{Q}[T]$, establishing the chain: $E_{P}[T] - \log E_{Q}[e^{T}] \leq E_{P}[T] - E_{Q}[T] \leq \sup_{f}\{ E_{P}[f] - E_{Q}[f] \} = D_{\mathrm{KL}}(P \| Q) = I(X;Y) $ , with supremum achieved at $f = \log \frac{dP}{dQ}$. The supremum $\sup_{T_\phi} \left( E_{P_{\text{joint}}}[T_\phi] - \log E_{P_{\text{marginal}}}[e^{T_\phi}] \right)$ thus recovers $I(X;Y)$ as $T_\phi$ converges to $\log \frac{P_{\text{joint}}}{P_{\text{marginal}}}$ at optimality, and proving $I(G^{\text{text}};G^{\text{image}}) \geq E_{P_{\text{joint}}}[T_\phi] - \log E_{P_{\text{marginal}}}[e^{T_\phi}]$.

---

> > ### Author Response · Authors · 2025-08-04
> >
> > **Question3: Does the network for mutual information estimation need to be trained for different datasets? Is this practical in FL considering the resource limitation and privacy concerns?**
> >
> > **Response3:**  **1)** The mutual information estimation network exhibits strong transferability across datasets sharing identical modality types (e.g., text-image → text-image) without requiring retraining, as it captures fundamental cross-modal dependency patterns rather than dataset-specific features. For transfers across divergent modality types (e.g., text-image → audio-video), a single retraining is recommended to adapt to new feature distributions, though the core architecture remains unchanged.  **2)** The lightweight MLP architecture (only 3-layer) ensures that the computational overhead for training the mutual information estimation network is negligible compared to the main MMFL training process. Crucially, all training runs locally on client devices - neither model parameters nor raw data are transmitted to the aggregation server or neighboring clients. This design guarantees both computational efficiency and strict privacy preservation, making the approach fully practical for FL deployments under resource constraints and privacy requirements.
> >
> > We sincerely appreciate the insightful suggestions and will implement a comprehensive restructuring to enhance methodological clarity and accessibility for general readers, particularly by refining the technical explanations in  Sections 3–4 and adding illustrative examples.

---

> ### Comment · Reviewer_AQiP · 2025-08-05
> **Further comments**
>
> I would like to thank the authors for their efforts in addressing my questions.
>
> Regarding the first question. The paper mainly considers approximate differential privacy according to Section 3.4  (your response above discusses pure DP, which differs from the manuscript and is confusing), and I assume the Gaussian mechanism is applied. I expect the authors to explicitly discuss how approximate DP is achieved, since many DP mechanisms in the literature guarantee approximate DP.
>
> The authors seem to utilize the basic composition theorem of approximate differential privacy, which renders $\epsilon$ for the overall privacy linear with respect to each modality. Then how about $\delta$? How large are $\delta$'s in the experiments? More importantly, the basic composition theorem is not tight enough, which makes the results less convincing.

---

> > ### Author Response · Authors · 2025-08-06
> >
> > We sincerely appreciate your comments. Our point-by-point response is presented below:
> >
> > **Question 1:** Regarding the first question. The paper mainly considers approximate differential privacy according to Section 3.4 (your response above discusses pure DP, which differs from the manuscript and is confusing), and I assume the Gaussian mechanism is applied. I expect the authors to explicitly discuss how approximate DP is achieved, since many DP mechanisms in the literature guarantee approximate DP.
> >
> > **Response 1:** We apologize for the earlier imprecision—our initial reference to pure DP was only intended to invoke the classic composition theorem and illustrate why $\sum_{i=1}^M \epsilon_i = \epsilon$ implies linear additivity of the privacy budget. In practice we use the Gaussian mechanism under $(\epsilon,\delta)$-DP, and thus the appropriate guarantee relies on the approximate-DP composition theorem, which states that composing $M$ mechanisms each satisfying $(\epsilon_i,\delta_i)$-DP yields overall $(\sum_{i=1}^M \epsilon_i,\sum_{i=1}^M \delta_i)$-DP. Concretely, for each modality $i$ we clip its gradient $g_i$ to $\|g_i\|\le C$  ($C$= 1 in our experiments) and add noise $n_i\sim\mathcal{N}(0,\sigma_i^2C^2I)$ where we set $\displaystyle\sigma_i=\frac{\sqrt{2Tln(1.25/\delta_i)}}{\epsilon_i}$ , and $T$ is the total number of communication rounds (following Wei Kang, et al. "Federated learning with differential privacy: Algorithms and performance analysis." *IEEE transactions on information forensics and security* 15 (2020): 3454-3469.).
> >
> > **Question 2:** Then how about $\delta$? How large are the $\delta_i$’s in the experiments?
> >
> > **Response 2:** In all experiments we set the total privacy failure probability to $\delta=10^{-5}$ (as described in line 261 in the paper), following common practice in the LDP literature, which offers a strong theoretical privacy guarantee while keeping the added noise magnitude moderate, and distribute it evenly across the $M$ modalities via $\delta_i=\delta/M$ (so $\sum_{i=1}^M\delta_i=\delta$), ensuring each modality mechanism satisfies $(\epsilon_i,\delta_i)$-DP and the composed mechanism satisfies $(\epsilon,\delta)$-DP.
> >
> > **Question 3:** The basic composition theorem is not tight enough, which makes the results less convincing.
> >
> > **Response 3:** We truly appreciate for raising this concern. We acknowledge that the basic composition theorem for $(\epsilon, \delta)$-DP—where privacy losses are summed linearly—is not the tightest possible bound, especially when many compositions are involved. However, in our current setting, we only apply the Gaussian mechanism to two modalities per instance, meaning that the privacy cost is composed over just $M = 2$ mechanisms. In this case, the basic composition yields a close approximation to the true privacy loss, and thus remains a justified simplification. That said, we fully agree that for more complex multimodal systems involving more modalities or longer training sequences, tighter composition tools—such as the advanced composition theorem, the Moments Accountant—could provide significantly improved bounds. We will add a note in Section 4.2 and the conclusion to clarify this limitation and highlight future directions involving tighter accounting frameworks.

---

> > ### Author Response · Authors · 2025-08-08
> >
> > We hope our response has addressed your concerns, and we would be more than happy to provide further clarification if needed.

---

### Official Review · Reviewer_X7LP · 2025-07-03

**Clarity:** 3
**Significance:** 3
**Originality:** 3
**Rating:** 4
**Confidence:** 3

**Summary:**

This paper proposes an information-theoretic approach to analyze privacy leakage in the process of MMFL. The proposed Sec-MMFL uses conditional mutual information to quantify the information leakage risk of each modality and dynamically adjusts the corresponding protection strength. Experiments on benchmark datasets demonstrate that Sec-MMFL outperforms traditional methods.

**Questions:**

The paper highlights that "different modalities require varying protection strengths" as a key motivation, but what is the specific solution and validation of this problem? Can conditional mutual information guarantee such a balance across modalities? More qualitative analyses on this issue would likely facilitate an intuitive understanding of the proposed model.

The current comparative methods do not adequately cover multimodal baselines. For instance, methods like LDP-Fed included in the comparisons are not designed for multimodal scenarios, making it difficult to effectively validate the advantages of the proposed method in MMFL settings. Although this issue is acknowledged in the limitations, mere textual mention cannot compensate for the lack of experimental evidence. This weakens the support for the claim that the proposed method concentrates on multimodal problems. It is recommended to supplement comparative experiments with mainstream multimodal methods to clarify the performance boundaries and advantages of the proposed approach.

The model framework (Figure 3) is hard to follow. It is claimed that "the Noise Scale Adjustment module adjusts the privacy budget assigned to each modality, achieving a better balance between privacy protection and model effectiveness," yet it does not seem to illustrate how this balance is achieved.

**Ethical Concerns:**

["NO or VERY MINOR ethics concerns only"]

**Final Justification:**

My questions have been fairly well addressed, and I choose to raise the score.

**Limitations:**

Yes

**Paper Formatting Concerns:**

There are no major formatting issues.

**Quality:**

3

**Strengths And Weaknesses:**

This paper focuses on the issue of MMFL and cross-modal leakage, which is an interesting topic. The proposed privacy-preserving approach is tailored to the characteristics of MMFL, with clear research objectives. However, the current comparative methods do not adequately cover multimodal baselines. The claimed solutions are not clearly validated or demonstrated qualitatively.

---

> ### Author Rebuttal · Authors · 2025-07-31
>
> **Question1: The paper highlights that "different modalities require varying protection strengths" as a key motivation, but what is the specific solution and validation of this problem? Can conditional mutual information guarantee such a balance across modalities? More qualitative analyses on this issue would likely facilitate an intuitive understanding of the proposed model.**
>
> **Response1:** Thank you for highlighting this crucial aspect. Our solution Sec-MMFL directly addresses modality-specific protection through two core innovations: **1)** Using conditional mutual information $I(G;D^i \mid D^{-i})$ (Eq. 8) to quantify the unique leakage risk per modality, accounting for inter-modal correlations. This allows precise risk assessment where text modality consistently shows higher risk than images, validating its sensitivity to modality characteristics. **2)** Adaptive noise allocation via Eq. 10 that inversely scales privacy budget $\epsilon_i$ with normalized risk $R'_i$ ensures high-risk modalities (e.g., text) receive stronger protection (lower $\epsilon_i$) while preserving utility for low-risk modalities. Experimental validation in Table 2 ($\epsilon=0.5$: TRR↓ 0.67 for text vs PSNR↓ 12.62 for images) demonstrates balanced protection. Qualitative analysis in Figure 5 visually confirms reduced cross-modal leakage, showing attackers cannot leverage recovered text to reconstruct high-fidelity images under Sec-MMFL, whereas uniform protection (LDP-Fed) allows clear image reconstruction when text is compromised. Additional parameter sensitivity analysis (Figure 4) further verifies stability across client counts and heterogeneity levels.
>
> **Question2/Weakness1: The current comparative methods do not adequately cover multimodal baselines. For instance, methods like LDP-Fed included in the comparisons are not designed for multimodal scenarios, making it difficult to effectively validate the advantages of the proposed method in MMFL settings.**
>
> **Response 2**: We appreciate this insightful suggestion. To strengthen validation, we have supplemented experiments with FedMultimodal [9], a state-of-the-art MMFL baseline, on CrisisMMD dataset, and Hateful Memes dataset. The table below presents a comprehensive comparison of all methods in terms of task performance and defense capability across two datasets. While FedMultimodal (specifically designed for MMFL) achieved marginally higher task accuracy on both datasets, it lacks dedicated defense mechanisms. In contrast, our Sec-MMFL method maintains comparable task performance while demonstrating significantly more balanced privacy protection across all modalities. We will add these results to Table 3 and explicitly discuss FedMultimodal comparisons in Section 5.3 of the revised manuscript. Furthermore, we will continue to incorporate additional mainstream multimodal methodologies to rigorously clarify the performance boundaries and comparative advantages of the proposed approach.
>
> | **Defense Method** | **CrisisMMD** | **CrisisMMD** | **CrisisMMD** | **Hateful Memes** | **Hateful Memes** | **Hateful Memes** |
> | ------------------ | ------------- | ------------- | ------------- | ----------------- | ----------------- | ----------------- |
> |                    | **Acc↑**      | **LPIPS↑**    | **TRR↓**      | **Acc↑**          | **LPIPS↑**        | **TRR↓**          |
> | LDP-Fed            | 0.652         | 0.583         | 0.80          | 0.572             | 0.663             | 0.75              |
> | NbAFL              | 0.635         | 0.602         | 0.82          | 0.564             | 0.670             | 0.73              |
> | LDP-FL             | 0.477         | 0.799         | 0.72          | 0.433             | 0.832             | 0.61              |
> | LDP-Fed-MI         | 0.643         | 0.756         | 0.64          | 0.569             | 0.891             | 0.57              |
> | FedMultimodal      | **0.658**     | 0.511         | 0.83          | **0.605**         | 0.622             | 0.78              |
> | Sec-MMFL           | 0.642         | **1.085**     | **0.53**      | 0.575             | **1.116**         | **0.49**          |
>
>
>
> **Question3: The model framework (Figure 3) is hard to follow. It is claimed that "the Noise Scale Adjustment module adjusts the privacy budget assigned to each modality, achieving a better balance between privacy protection and model effectiveness," yet it does not seem to illustrate how this balance is achieved.**
>
> **Response 3**: Thank you for this valuable feedback. We will revise Figure 3 to explicitly visualize the balancing mechanism: 1) Add directional arrows showing the flow from Leakage Risk Estimator (outputting $R_i)$ to Noise Scale Adjustment, where Eq. 9-10 convert $R_i$ into $\epsilon_i$ values (e.g., text: \($\epsilon_{\text{text}}$=0.2$\epsilon$), image: $(\epsilon_{\text{image}}=0.8\epsilon)$ when $(R_{\text{text}} > R_{\text{image}}$)). 2) Include gradient icons with variable noise magnitudes (larger noise symbols for text, smaller for images).  This clarifies how high-risk modalities induce lower $\epsilon_i$ (stronger noise), while low-risk modalities retain higher $\epsilon_i$ (weaker noise) - precisely balancing protection and utility. The empirical evidence of this balance is demonstrated in Table 2 (e.g., at $\epsilon=1$, text TRR improves by 26% and image PSNR degrades only 6% vs LDP-Fed).

---

> > ### Comment · Reviewer_X7LP · 2025-08-05
> >
> > A new experimental result of FedMultimodal (2023) is provided in the response. I have observed that this comparison was previously mentioned by another reviewer. It is OK. But does this mean that even after adding this new comparison, the entire work only includes one comparison with the multimodal method? Theoretically, including only one comparison method of the same type would indeed weaken the persuasiveness of the comparative analysis. It is possible that there truly are no more methods of the same type available for comparison in this field. In such cases, it would be helpful if the authors could provide a more detailed explanation to clarify this situation.

---

> > > ### Author Response · Authors · 2025-08-07
> > >
> > > It is worth noting that multimodal learning has become a rapidly emerging and highly popular research area only in recent years. Consequently, the attention to privacy-preserving defenses tailored specifically for multimodal federated learning has been relatively limited until now. Despite the current scarcity of comparative works, the importance and broad applicability of multimodal distributed training and learning systems have grown substantially, making the development of corresponding defense mechanisms a crucial and timely research frontier. Moreover, designing effective privacy defenses for multimodal data presents significant intrinsic challenges due to the heterogeneous nature of multiple data modalities and the complexity of their joint distributions. Compared to more straightforward unimodal scenarios, multimodal privacy protection requires sophisticated methodologies that can balance utility and privacy across diverse data types and fusion architectures. This inherent difficulty partially explains the limited number of existing multimodal federated learning defense frameworks.
> > >
> > > Therefore, our work can be seen as an initial effort to address these complex challenges at the emerging frontier of privacy-preserving multimodal federated learning. The limited availability of directly comparable baselines reflects the novelty and technical complexity of this area, rather than any shortcoming in our approach. We expect that our findings will offer valuable insights and a basis for further research in this area.

---

> ### Author Response · Authors · 2025-08-07
>
> **Response:** We sincerely appreciate your valuable feedback regarding the sufficiency of multimodal method comparisons. We fully acknowledge the concern that supplementing only FedMultimodal as a comparative baseline might appear limited. This situation reflects a fundamental research gap in the MMFL field: while numerous studies focus on improving multimodal fusion accuracy (e.g., [9,25,45] in our references), very few have explored dedicated privacy preservation mechanisms tailored for multimodal scenarios.  Several recent works have introduced novel defense mechanisms targeting entirely new threat scenarios without having directly comparable prior baselines. For example, SafeSplit (NDSS 2025) is the first defense designed for client‑side backdoor attacks in Split Learning. Existing defenses from other distributed frameworks like Federated Learning are not applicable, and there is a lack of effective backdoor defenses specifically designed for Split Learning. Therefore, they adopt conventional unified defense strategies like KRUM against backdoor attacks and apply them to the Split Learning setting as baselines for comparison. Similar baseline setting strategies are also adopted in works such as Dual Defense (NeurIPS 2024). In studies exploring differential privacy mechanisms in new scenarios, it is common to directly incorporate standard differential privacy as a comparative baseline, as in NetDPSyn (IMC2024). As the research on defense methods under MMFL is still in its early stage and lacks established baselines, we follow a similar comparative evaluation setup in our work.
>
> To address this limitation comprehensively, we have conducted substantial additional experiments supplementing two state-of-the-art MMFL frameworks - FedSea: Federated Learning via Selective Feature Alignment for Non-IID Multimodal Data ( IEEE TRANSACTIONS ONMULTIMEDIA 2023)  and HAMFL: Adaptive Hyper-graph Aggregation for Modality-Agnostic Federated Learning (CVPR 2024) . Both were adapted with equivalent basic differential privacy mechanisms (δ=1e-5, sensitivity=1.0) under identical attack scenarios to ensure fair comparison. The complete experimental results now demonstrate consistent trends across five privacy methods and three MMFL-specific baselines:
>
> | **Defense Method** | **CrisisMMD** | **CrisisMMD** | **CrisisMMD** | **Hateful Memes** | **Hateful Memes** | **Hateful Memes** |
> | ------------------ | ------------- | ------------- | ------------- | ----------------- | ----------------- | ----------------- |
> |                    | **Acc↑**      | **LPIPS↑**    | **TRR↓**      | **Acc↑**          | **LPIPS↑**        | **TRR↓**          |
> | LDP-Fed            | 0.652         | 0.583         | 0.80          | 0.572             | 0.663             | 0.75              |
> | NbAFL              | 0.635         | 0.602         | 0.82          | 0.564             | 0.670             | 0.73              |
> | LDP-FL             | 0.477         | 0.799         | 0.72          | 0.433             | 0.832             | 0.61              |
> | LDP-Fed-MI         | 0.643         | 0.756         | 0.64          | 0.569             | 0.891             | 0.57              |
> | FedMultimodal      | 0.658         | 0.511         | 0.83          | 0.605             | 0.622             | 0.78              |
> | FedSea             | **0.665**     | 0.497         | 0.79          | 0.592             | 0.635             | 0.76              |
> | HAMFL              | 0.661         | 0.523         | 0.81          | **0.612**         | 0.641             | 0.77              |
> | Sec-MMFL           | 0.642         | **1.085**     | **0.53**      | 0.575             | **1.116**         | **0.49**          |
>
> These extended results reveal three patterns: First, all MMFL+DP variants (FedMultimodal, FedSea, HAMFL) exhibit consistently high text leakage, confirming that even advanced multimodal architectures remain fundamentally vulnerable to gradient inversion attacks when using generic protection. Second, Sec-MMFL achieves  higher LPIPS scores, indicating stronger protection for visual content - particularly crucial given images' higher sensitivity. Third, while specialized MMFL methods show marginally better accuracy, they do not have an advantage in terms of a more balanced level of privacy protection.

---

> > ### Comment · Reviewer_X7LP · 2025-08-07
> >
> > Thanks for the authors' careful responses. My concerns have been well addressed.

---

> ### Author Response · Authors · 2025-08-08
>
> We sincerely appreciate your recognition, and your feedback has been invaluable in improving the clarity and quality of our work.

---

### Note · Authors · 2025-08-16

We deeply appreciate the reviewers for your rigorous evaluation and constructive feedback. We’re particularly grateful for recognizing the novelty and pioneering nature of our work in addressing privacy challenges specific to multimodal federated learning, a key gap in existing research. Your acknowledgment of our information theory based framework, which uses conditional mutual information to quantify modality-specific risks, as an innovative approach means a lot to us.

 To address your concerns , we’ve made the following improvements:

- For multimodal baseline comparisons (Reviewers X7LP, 3U4G), we added experiments with FedMultimodal, FedSea, and HAMFL (all with equivalent differential privacy). Results show Sec-MMFL outperforms these methods in balanced protection—reducing text leakage and boosting visual security  while keeping utility steady.
- To strengthen mathematical rigor (Reviewer AQiP), we corrected Equation (10) and added proof that the privacy budget allocation guarantees total $(\varepsilon, \delta)$-DP via approximate composition. We detailed the Gaussian mechanism implementation and clarified how we use MINE for mutual information estimation.  Furthermore, we have elaborated on the neural estimation workflow for cross-modal mutual information, clarifying how gradient tensors are flattened, concatenated, and processed via a Donsker-Varadhan optimized critic network to handle architectural heterogeneity.
-  We expanded experiments (Reviewers AQiP) to include state-of-the-art attacks (e.g., "See through gradients" [40]) which further confirm our method’s robustness across different settings.
-  Regarding noise profile leakage (Reviewer EEh7): attackers only access perturbed gradients, and LDP ensures irreversibility—preventing exploitation of modality risk meta-information.

We also noticed a formatting issue in our response to Reviewer AQiP: due to markdown compilation limits on web pages, the gradient clipping threshold (intended to be the $\ell_2$ norm $||g_i|| \leq C$) was incorrectly displayed as the $\ell_1$ norm $|g_i| \leq C$. We sincerely apologize for this mistake.

Your thoughtful feedback has been instrumental in refining our work, and we’re grateful for the opportunity to strengthen its rigor and clarity while advancing this important area of research. Thank you for your time and consideration of our submission.

Sincerely,

Authors of  submission 16629

---

### Decision · Program_Chairs · 2025-09-17

**Decision:**

Accept (poster)

**Comment:**

This paper proposes a method for securing multimodal federated learning via differential privacy. Reviewers praised the methodological development and empirical evaluations in the paper, with the major technical concern raised by two reviewers of a lack of comparison with dedicated multimodal baselines being successfully addressed.

After the rebuttal and discussion, the remaining negative reviewer bases their evaluation largely on an unclear explanation of the differential privacy details and other clarity issues. These concerns, which I have copied below for completeness, were either not shared or viewed as not so significant by the other reviewers. In my estimation, I agree that the clarity issues are not so severe as cannot be fixed in a revision, and so I am tentatively recommending acceptance. However, I encourage the authors to carefully consider and revise according to the clarity suggestions given by the reviewers.

## Reviewer AQiP's discussion comments:

My major concern about this paper is that the authors seem not familiar with the differential privacy literature. For example, they utilized $\sigma_i\propto\sqrt{T\ln(1/\delta)}/\epsilon$, which is a result of moments accountant for aggregating privacy over $T$ communication rounds (and somehow the privacy amplification effect of subsampling is ignored, see the seminar work Abadi, et al., Deep learning with differential privacy). The authors referred to Wei Kang, et al, but the reference considered a different context. However, when composing privacy over modality, they use the basic composition theorem. Limiting the results to only 2 modalities may also hinder the applicability of the proposed method.

The writing in manuscript and response lacks some rigor as well, for example, in their response, they claimed a clipping threshold of $|\mathbf{g}_i|<C$, but $\mathbf{g}_i$ is a vector, and the Gaussian mechanism assumes $|\|\mathbf{g}_i|\|_2<C$ (note that different mechanisms may require different types of norms for clipping. The Laplace mechanism requires $L_1$-norm).

Overall, I feel this paper was prepared in a rush, and a major revamp will significantly improve the clarity of the paper. Without another round of careful review, it is not certain if the paper will be clear enough for a publication in my opinion.